# Biomechanical Hand Prosthesis Design †

Emilia Furdu Lunguţ [1], Lucian Matei [2], Maria Magdalena Roşu [3] , Mihaiela Iliescu [4,*] and Corina Radu (Frenţ) [4,*]

1   CF2 Hospital, Titu Maiorescu University, Mărăşti 63, 011464 Bucharest, Romania; furduemilia@gmail.com
2   Department of Automotive and Industrial Engineering, Faculty of Mechanics, University of Craiova, 107 Calea Bucuresti Street, 200512 Craiova, Romania; lucian.matei@edu.ucv.ro
3   Faculty of Industrial Engineering and Robotics, National University of Science and Technology Politehnica of Bucharest, Splaiul Independentei 313, 060042 Bucharest, Romania; magdalena.rosu@upb.ro
4   Institute of Solid Mechanics, Romanian Academy, Constantin Mille 15, 010141 Bucharest, Romania
*   Correspondence: mihaiela.iliescu@imsar.ro (M.I.); corinagabrielaradu@imsar.ro (C.R.)
†   This paper is an extended version of our paper published in Todiriţe, I.; Radu Frenţ, C.; Iliescu, M. Basics of Hand Prosthesis Design. In Proceedings of the 8th International Workshop on New Trends in Medical and Service Robotics (MESROB 2023), Craiova, Romania, 7–10 June 2023; pp. 379–385.

**Abstract:** There are various studies on the structural and functional constructions of hand prostheses inspired by human biomechanics, and there are different kinds of prostheses available on the market. This paper aims to present the relevant stages of designing a hand prosthesis prototype that is innovative due to its mechanical structure and, therefore, the prosthesis fingers' DOF and mobility. The prosthesis is designed to have independent finger motion with the rotations of each of the three phalanges and, most importantly, rotation for each of the fingers relative to the palm. All these motions are generated and controlled by micromotors, a microcontroller, and sensors. A reverse engineering technique was applied for obtaining the exterior surface dimensions of the prosthesis and this consequently ensures that this prosthesis looks as realistic as possible. Small, light mechanical parts were designed as components of the mechanical system for the motions of finger phalanges and most of them (gears, levers, shells) were made using 3D-printing technologies (digital light processing (DLP) and/or selective laser sintering (SLS)). Aspects of some technical problems which arose during the prototype assembly are also recorded in the paper. Further research development will focus on the tests conducted on the prosthesis and the consequent adjustments of the prototype.

**Keywords:** hand prosthesis; kinematic analysis; reverse engineering; 3D model; rapid prototype





## 1. Introduction

The human body is a very special and complex system that should ideally function and work almost perfectly all throughout a person's life. Still, there are situations when something can go awry and one or more subsystems (parts of the human body) become damaged, so external intervention is required to reduce suffering and to improve quality of life.

The limbs are beneficial to everyday functioning for humans, whether the activity involves motion and posture or focuses on performing tasks for work and everyday life. If any part of a limb is missing, serious problems arise, and efforts can be undertaken to replace the missing part.

Knowing the biomechanics of humans' upper limbs is crucial for the design of a hand prosthesis. The authors of [1] present a detailed study on this topic, aiming to help researchers develop their knowledge of the body's structure and postural functionality and of therapy for the medical recovery of hand movements.

The study in [2] analyses, from a biomechanical point of view, the way in which certain activities are carried out, for example, the actions of people with upper limb motor impairments on equipment touch screens. The authors noted that the entire chain of

segments and joints must be correlated to achieve the movements but also that the shapes of the devices can be modified in order to increase ease of use.

Mechanically, hand prostheses could be assimilated to a chain of mechanisms that transmit motion by actuators/motors, gears, and levers from muscle or cortex sensors toward the fingertips of the prosthesis.

Prosthetic components include the socket, suspension, control system(s), joints, and appendages. There are many different options for prostheses, all aiming to achieve stable, comfortable, and optimum function.

A prototype for an affordable mechatronic system which is tailored to the needs of upper limb amputees is presented in [3]. The study presents a pneumatic arm actuation that could be used in precision activities, such as surgery. The results of weight-lifting experiments using these prostheses showed increased flexibility and reduced reaction time compared to hydraulically actuated systems.

The design of human wrist prostheses with three degrees of freedom based on an asymmetric parallel mechanism is presented in [4]. The results shed light on both inverse kinematic analysis and velocity mobility analysis. Experiments were carried out to prove the prosthesis's ability to perform sequences of movements similar to those of the human hand.

The collection of the skin's EMG signals is carried out with the help of sEMG sensors, which sense the electrical signals generated by the healthy muscles of the arm and forearm. These signals are detected by sensors placed on the surface of the skin and are then transmitted to the microprocessor of the prosthesis, which interprets the signal and triggers the movements of the artificial hand.

Myoelectric prostheses for the hand are made of electronic and mechanical components, as well as components that ensure an acceptable aesthetic appearance. The electronic components include sensors, amplifiers, processors, and batteries. The electrical signals generated by the user's healthy muscles are amplified to control the motors. The control unit is responsible for detecting electrical signals and decoding them into the movements performed by the prosthesis [5].

Smart bionic hand prostheses are made of modern, waterproof, and flexible materials, motion sensors, electric motors, control algorithms, and a user interface that allows them to recognize and interpret the electromyography (EMG) signals generated by the healthy muscles from the user's arm. The fingers of this prosthetic hand are made in such a way as to reproduce natural movement; a simple mechanism with four bars that move four fingers is implemented. The four-bar link is the simplest closed movable link with only one degree of freedom. It has four components: crank, coupling, rocker, and frame [6].

There are various studies on the structural and functional constructions of hand prostheses [7,8] inspired by human biomechanics and focused on postural functionality, medical recovery, or smart prostheses design for amputated limbs. In [2], for example, researchers approached the task from a biomechanical point of view and presented results on the way people with upper limb motor impairments interact with the touchscreens of different equipment.

The authors of [2] present a comparative study on grip modes and grip forces performed by two groups, one among people not wearing upper limb prostheses and the other among people wearing upper limb prostheses. The results of this study were used for further improving force control strategies and for developing a new method of adjusting the prosthesis to the limb abutment that is customized for each person.

The comfort and ergonomics of use, as well as durability and production costs, are taken into account. Specifically, the quantified assessment of anthropomorphism refers to the assessment of the degree to which the prosthetic hand is perceived as human. This assessment can be performed with questionnaires or psychological tests. The results of the questionnaires measure a user's level of comfort and confidence in using a prosthetic hand.

The optimization and finalization stage of a prosthetic hand involves adjusting and improving the design and functionality so it will meet a user's requirements and needs.

Nowadays, on the market, there are different models of prosthetic hands available. Manufacturing companies are still performing research studies to improve them.

The authors of [9] evidence the concept, design, and results for an anthropomorphic prosthetic hand with five fingers that performs everyday grasping activities. The motions control is myoelectric and real-time, so there are two sensors used. The mechanical structure is enabled by four actuated motors so that four degrees of freedom are available for the fingers and it is possible to express eight hand specific gestures.

The authors of [10] present a male prosthetic hand with ten degrees of freedom which was designed, 3D-printed, and tested in a real environment. The mechatronic system includes a video camera in the palm, enabling perception of the surrounding environment, and a display on the back of the hand. The low-cost fabricated prototype has five fingers and two motors, providing good mobility and extremely fast postural closure (about 1.30 s).

A biomechanically suitable upper limb prosthesis for the human wrist is presented in [11]. The paper presents an anthropomorphic modular structure with two degrees of freedom that enable human-like movements with reasonable performance in grasping essential objects.

The design and the prototype of an under-actuated anthropomorphic prosthetic hand system are shown in [12]. It is a five-finger prosthesis that enables human-like grasping movements due to a high-performance multisensory electromyographic system and a six-motor-drive system. The phalanges and the finger joints position control is based on the information feedback from vibratory, sound, and visual sensors.

The prosthetic hand mentioned in [5] benefits from a new hybrid control model. It enables a user to influence the grasping mode and grasping force through an intelligent system. The electrical activity of the muscles is registered by the system and further analysed through a consideration of the voltage and the duration between stimuli. The obtained results prove that the system works efficiently.

With the aim of attaining a low cost of manufacture and relatively quick production, the authors of [5] present the design and prototype of a modular prosthetic hand which incorporates a shape memory actuator for the index finger. Research results on three different types of elastic materials for injection mould gloves are presented in [13]. The materials envisaged for this product are silicone rubber, polyvinyl chloride, and thermoplastic styrene elastomer. Silicone rubber is the most suitable for sensor mounting and thermoplastic styrene elastomer has the best mechanical properties and the best appearance.

The aspects mentioned above, as well as many of the research papers and market-available products studied by the authors, prove that there is a lot of interest in prostheses for missing limbs—with hand prosthesis being the most relevant. There are different structures and designs with various numbers of DOFs available, consequently with different motions and tasks that could be performed. Command and control systems are mainly based on electromyography, and sensory processing AI has also been considered lately. Most of the prototype manufacturing techniques point toward additive manufacturing, namely rapid prototyping.

This paper aims to present the relevant stages of designing a hand prosthesis prototype that is innovative in its mechanical structure, in the DOFs of the fingers, and in its mobility. The customized elements in manufacturing and in the preliminary testing phases of the prototype are also discussed and shown in the relevant sections of the paper. The contents of the sections are as follows: Methods and materials—points out the concept, design, kinematic analysis, and reverse engineering techniques applied (command and control method; 3D printing materials). Results and Discussion—evidences research results and discussion on their significance. Conclusion—focuses on achievements and future developments of the research.

## 2. Methods and Materials

The impetus for designing a hand prosthesis (which was different from the ones available on the market) came as a special request from the General Directorate of Social

Assistance and Child Protection (from a Romanian county). One of the teenagers living in foster care had experienced an accident and had lost her hand. The only affordable prosthesis was an esthetical one with low functionality.

The idea of manufacturing an individualized hand prosthesis with relatively high functionality at an affordable price came in response to the need among people with disabilities, especially people with a missing limb and low income, to attain improved quality of life.

Upper limb motility comprises complex functions which are made possible through the interaction of complex nervous systems: the pyramidal motor, the extrapyramidal motor, and the peripherical motor. The direct pathway for movement occurs through the transmission of voluntary motor input; the indirect pathway for movement occurs through the transmission of automatic or semi-voluntary motor input and contributes to the regulation of muscle tone. The motor unit, as a functional unit, is made of motoneuron, axon, and the muscle fibres which are innervated by it.

The prosthesis presented in this paper was designed to have independent finger motions that stand in rotations for each of the three phalanges and, most importantly, have rotation for each of the fingers relative to the palm. All these motions are generated and controlled by sensors, micromotors, and a microcontroller. The prosthesis was designed to have a low weight, to be user-friendly, and to be financially affordable. The design of the upper limb prosthesis is shown in Figure 1. One can notice the initial concept for the whole prosthesis—including the hand, the forearm, and the arm—will be fixed on the residual limb (abutment). Figure 2 points toward the final concept of the hand, evidencing the mechanical parts: the levers system, the worm–wheel gears, the bearings, and the micromotors for the phalanges' motion.

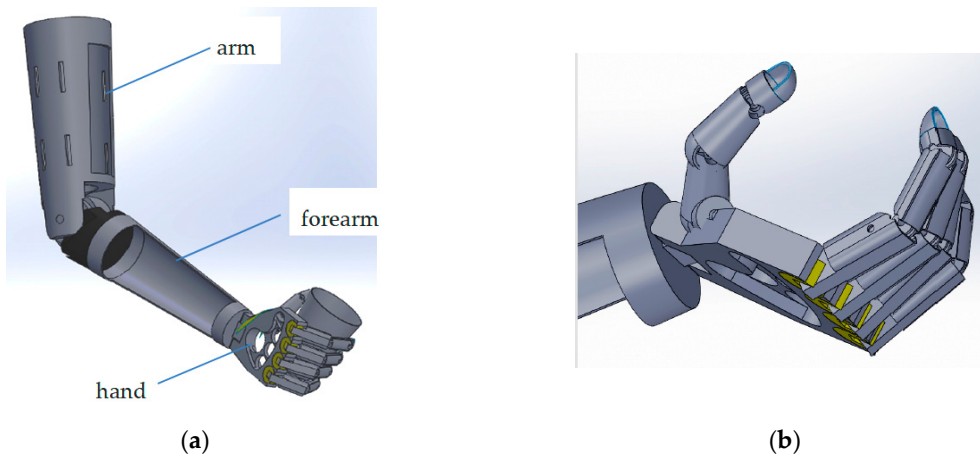

(**a**)                                                                 (**b**)

**Figure 1.** Upper limb prosthesis concept: (**a**) prosthesis; (**b**) hand.

The concept and design for the here-presented hand prosthesis are innovative and some of the relevant aspects are described here:

- Each of the phalanges has the potential for rotation and there is additional rotation for the whole finger;
- Each of the fingers possesses independent motion;
- Solely electrically driven finger which is customized by both its soft and hard structures;
- Motion is achieved by levers and gears (spur, worm) with no cables and elastic elements (spring);
- Command and control is performed through micromotors (not servomotors or encoders), a microcontroller, and the sensors' signals.
- Sensor types include pressure, inclination, temperature, gyroscope, and sEMG.

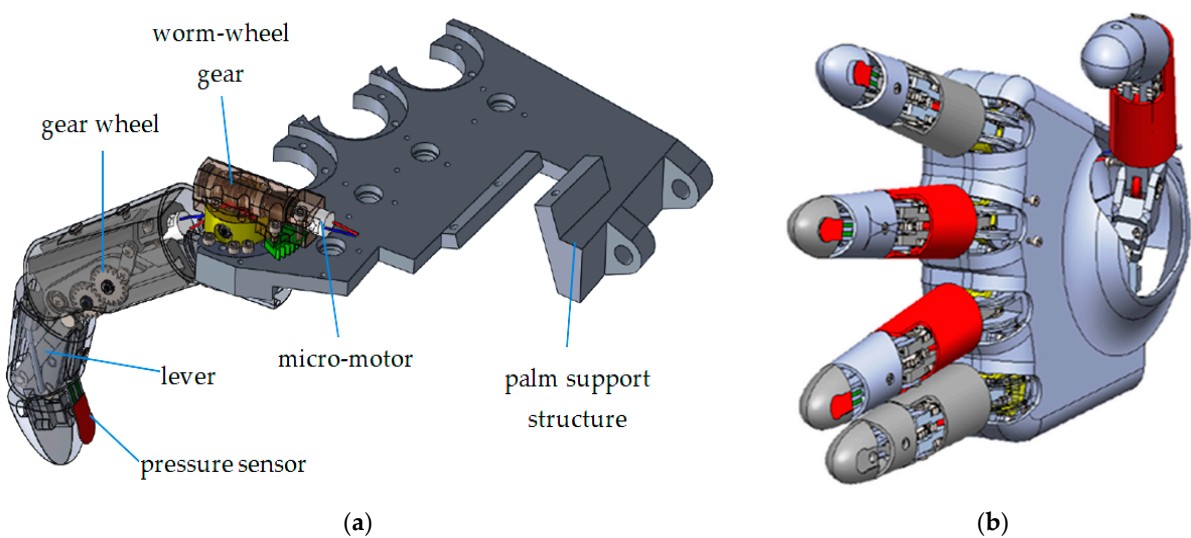

**Figure 2.** Hand prosthesis design: (**a**) main components; (**b**) 3D model.

The concept of the prosthesis's finger is that, of the three phalanges, the first one, P1, (metacarpus–phalange) has two independent rotational motions. The other two phalanges (intermediate (P2) and fingertip (P3)) have dependent rotations, enabled by a levers mechanism (see Figure 3). The T notation is for the fingertip.

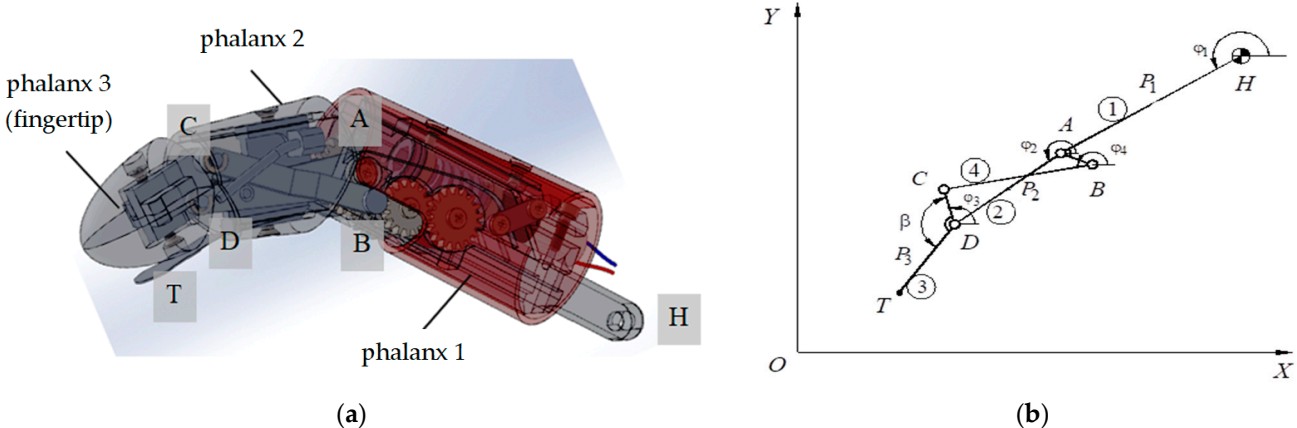

**Figure 3.** Hand prosthesis's finger concept: (**a**) 3D model; (**b**) kinematic scheme.

Classical notations for the mechanisms are performed in letter notation for the joints and number notation for the kinematic elements. Based on this assumption, considering the relative motions of the elements, the finger mechanism is made of lower pair joints: H(0R1), A(1R2), B(1R4), C(3R4) and D(2R3). Here, the notation R stands for the rotational motion.

The mechanism has four mobile elements: 1(A,H,B), 2(A,D), 3(C,D), and 4(B,C).

Based on the above-mentioned numbers of mobile elements and kinematic joints, the resulting degree of mobility for the mechanism is M = 2.

The block (multipolar) scheme of the finger mechanism is presented in Figure 4, and is made of the following components: base Z(0), motor groups R(1) and R(2), and the triad (RRR)(3,4). Due to the above-mentioned aspects, the structural relation of this mechanism results in:

$$Z(0) + R(1) + R(2) + RRR(3,4)$$

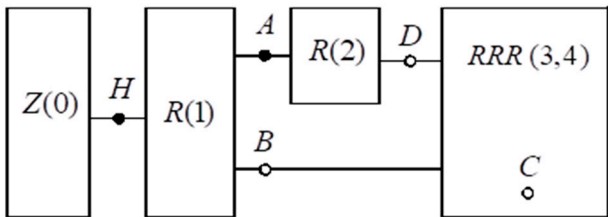

**Figure 4.** Hand prosthesis's finger block scheme.

A kinematic analysis of a mechanism involves determining the position, speed, and acceleration of each of its mobile elements.

For the studied mechanism, the analysis was performed using special procedures, namely A1R_G, A1RALFA_G, and D1PVA_G [14,15]. These procedures consist of forward kinematic analyses based on a vector loop method. Position, speed, and acceleration calculations were performed for 26 equidistant positions of element 1, considering the initial position: $\varphi_1 = \varphi_{10} = 3.2$ rad.

The sequence of the kinematic analysis main program lines are shown in Figure 5.

```
% Dimensions
 ah = 0.0531; ad = 0.0284; ab = 0.0048; bc = 0.0275;
 cd = 0.00675; dt = 0.020;

% constructive data
 H = [0.12 0.045 0 0 0 0];
 H1 = [0.12 0.055 0 0 0 0];
 fi1 = [3.2 0.1 0]; fi2 = [3.3 0.2 0]; fi3 = [2 0 0];
 fi4 = [2.8 0 0];

 fi10 = 3.2; fi20 = 3.3; alfa = 1; beta = 2.303834;
 pas = pi/180;
 r = 0.001; r1 = 0.005; r2 = 0.003;
```

```
for i = 1 : 26
    fi1(1) = fi10+(i-1)*pas;
    fi2(1) = fi20+3*(i-1)*pas;
    A = A1R_G(H,fi1,ah);
    B = A1RALFA_G(A,fi1,ab,pi-alfa); hold;
    D=A1R_G(A,fi2,ad); hold;
    [fi3, fi4, C] = d1pva_G(D,B,fi3,fi4,cd,bc); hold;
    T = A1RALFA_G_T(D,fi3,dt,beta);
    CB(H,r);

    XT(i) = T(1);   YT(i) = T(2);
    plot(XT,YT,'o','LineWidth',1.5,'MarkerEdgeColor','k',...
        'MarkerFaceColor','w','MarkerSize',1.5);
    XD(i) = D(1);   YD(i) = D(2);
    plot(XD,YD,'o','LineWidth',1.5,'MarkerEdgeColor','k',...
        'MarkerFaceColor','w','MarkerSize',1.5);
    XA(i) = A(1);   YA(i) = A(2);
    plot(XA,YA,'o','LineWidth',1.5,'MarkerEdgeColor','k',...
        'MarkerFaceColor','w','MarkerSize',1.5);

    fi11(i)=fi1(1); fi21(i)=fi2(1); fi31(i)=fi3(1)+beta;
    f1v(i) = fi1(2); f2v(i) = fi2(2); f3v(i) = fi3(2);
    f1a(i) = fi1(3); f2a(i) = fi2(3); f3a(i) = fi3(3);
    i1(i) = i;
%
    CB(H1,r);
```

**Figure 5.** Kinematic analysis program lines.

For the prosthesis, its exterior surface design was determined using a reverse engineering technique, applied to an existing arm (Figure 6) of the person the prosthesis was being made for. The equipment used was a MetraSCAN 3D scanner [16]; the scanning process applied a mesh resolution of 0.1 mm with 800,000 measurements/s. Once the surface was generated, its dimensions in any plane and sections were determined by slicing it with the corresponding planes.

The hand prosthesis is a complex system with embedded command and control functions (see Figure 7). The sensor which receives the command signal is an sEMG (surface electromyography) that is intended to be used directly, providing an amplified and integrated signal to be used by the microcomputer that contains the analogue-to-digital converter (ADC).

The sensors that enable the acquisition of the data for prosthesis functioning are the pressure and tilt angle sensors. These sensors are used by the ADC (analog-to-digital converter) system of the microcomputer. Their structure is a resistive one in the form of a layered film.

The microcomputer which was used for initial tests was an ATSAM3X8EA (produced by Microchip/Atmel and distributed by Mouser, https://eu.mouser.com/); the motor driver was a DRV8711 (produced by Texas Instruments and distributed by Farnell, https://uk.farnell.com/), and the programming language used was Python 3.7. Each finger of the prosthesis is driven by three independent micromotors, resulting a total of fifteen for all the fingers of the hand.

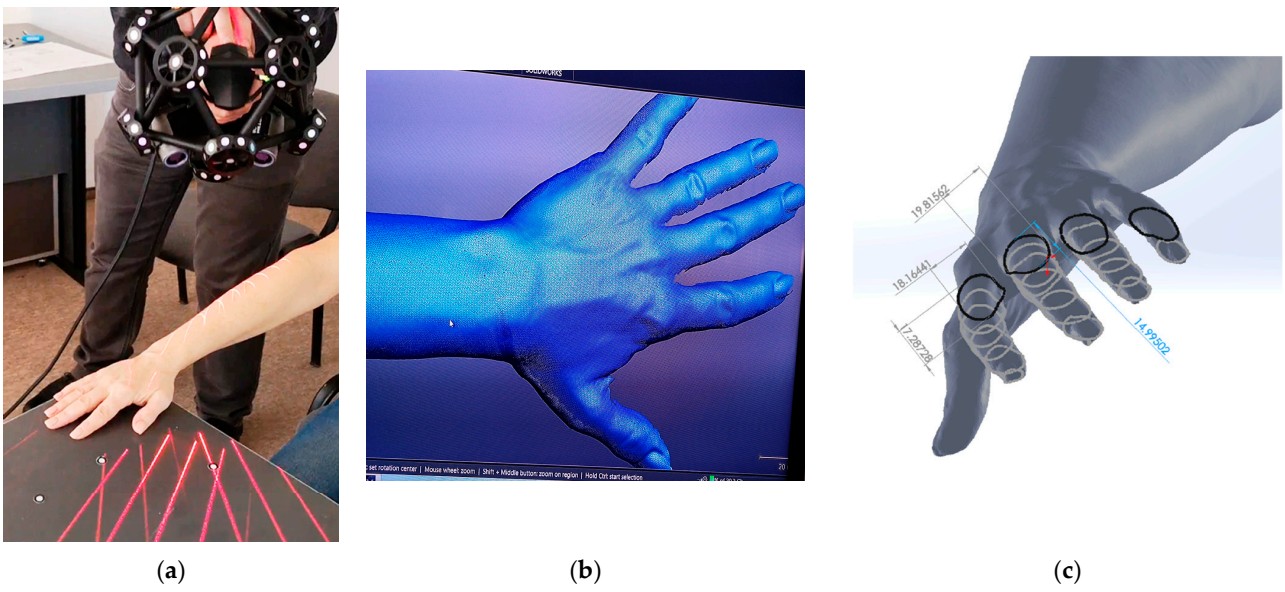

**Figure 6.** Reverse engineering applied for prosthesis exterior surface design: (**a**) 3D scanning; (**b**) scan of hand surface; (**c**) fingers section dimensions.

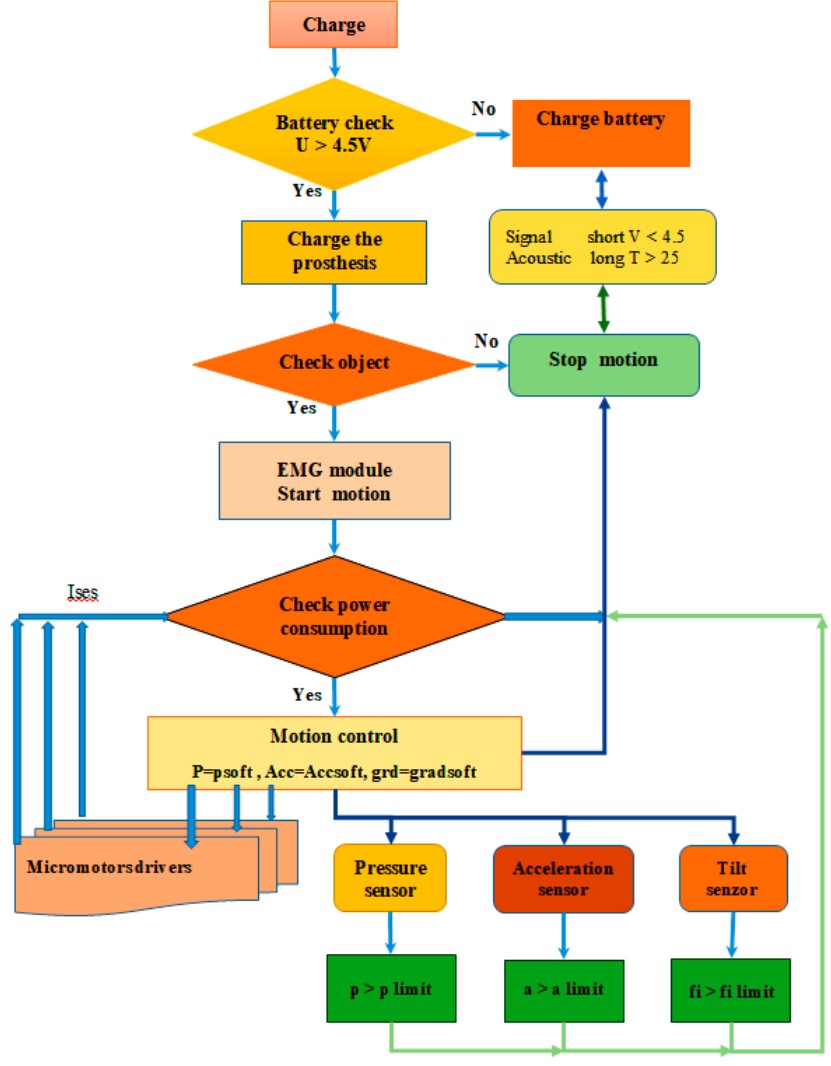

**Figure 7.** Hand prosthesis logical schema.

Basically, each of the horizontal and vertical axes motions of the fingers has independent command and control due to the innovative design of the prosthesis.

The command and control system is made of subsystems, including the sensory subsystem (pressure, inclination, temperature, gyroscope, surface EMG); the driving subsystem (15 micromotors); the control subsystem (microcomputer, micromotor controllers, GPIO, and power expanders) (see Figure 8).

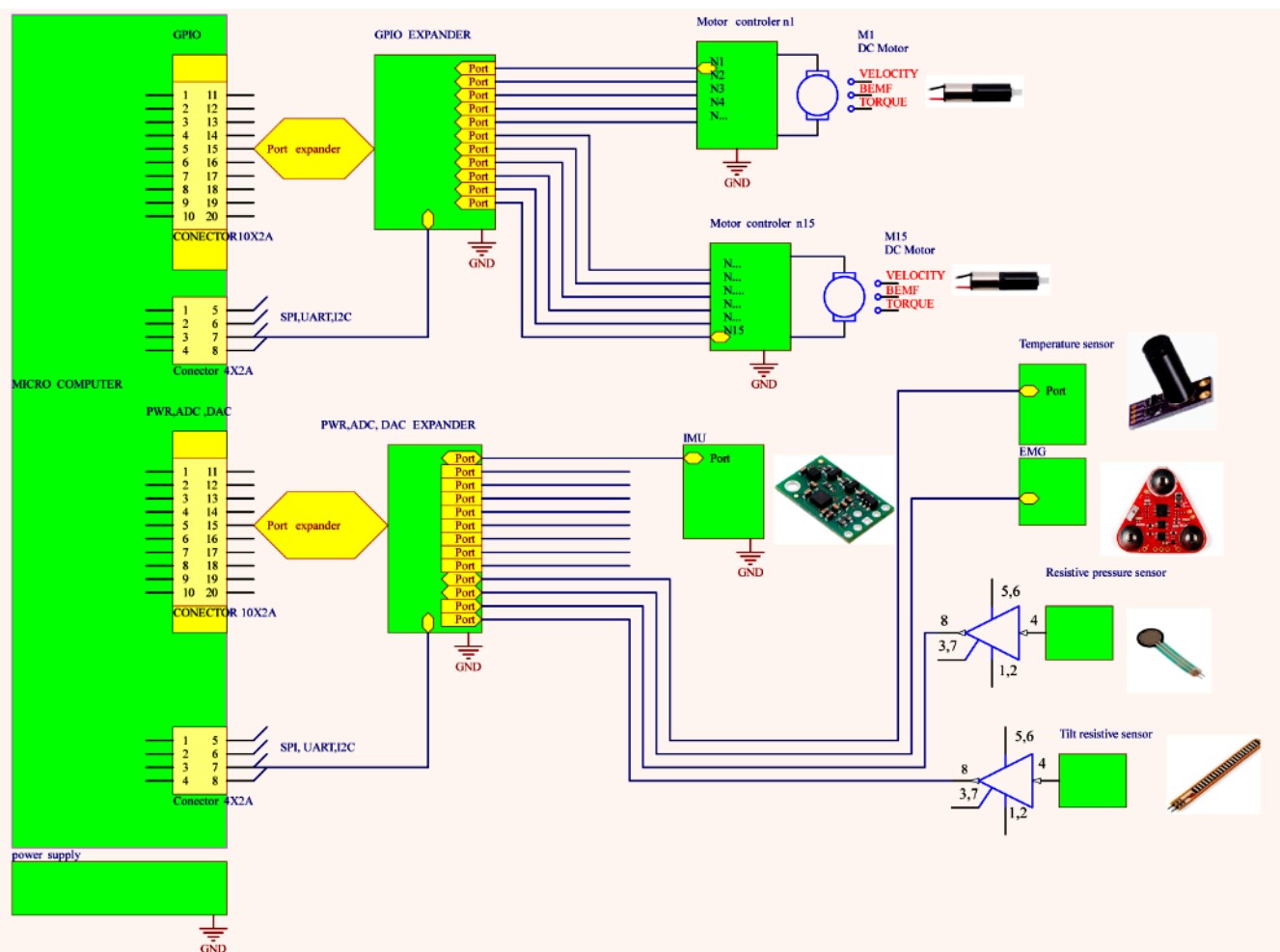

**Figure 8.** Command and control components scheme.

Last but not least, a questionnaire was created for market research purposes to determine the need and requirements for prostheses as expressed by persons with upper limb disabilities. The questionnaire is available on Google forms (Chestionar SISTEM BIOMECANIC PENTRU MEMBRU SUPERIOR UMAN). The questionnaire was completed by more than 50 people. A further, detailed survey was carried out; this was designed to determine the specific needs of the people who agreed to participate in the survey.

The basic ideas for prototyping the hand prosthesis were based on the available materials and different technologies that could produce a reliable and sustainable product, while having an affordable cost and requiring low maintenance.

Considering the many available 3D-printing technologies, we applied the ones which were available within the authors' research facilities and, importantly, those which were financially affordable. We tested various 3D-printing techniques, but not all of them rendered parts with satisfactory quality. The results of the tested technologies are presented here:

- The first rapid prototyping technology tested was an FDM (MarkTwo 3D Printer from Markforged). The prototype has multiple small parts (for example the 0.5 mm thread

pitch), and the technological limitations of the printer could not provide the geometric details needed.

- The second rapid prototyping technology tested was an SLA (Form 2 from FormLabs), but because of the technological limitations of the printer, this technique did not prove to be adequate. These limitations included the following: laser spot size for XY (horizontal) plane; available layer height (minimum of 0.03 mm); long print duration at the minimum layer height.
- The third rapid prototyping technology tested was an MJF (Jet Fusion 5200 from HP). The printed parts, as green parts, were in accordance with the geometrical precision required from the point of view of the material and its resistance strength, but the post-processing (sandblasting) stage caused failures in the geometric accuracy of the printed thread (for example).

Based on all the above-mentioned findings and, importantly, the findings of the "screening" process of the available prototyping techniques (those presently available to the authors), the decision was made to apply and test two other types of rapid prototyping technologies: digital light processing (DLP) [16,17] and selective laser sintering (SLS) [18,19].

As previously mentioned, the main problem that arose in relation to the different tests of the 3D-printing techniques was that of the geometric precision of the printed parts; for example, the pitch for the worm should be 0.5 mm and the module for the gear should be 0.6 mm. These are parts which require high geometric precision.

DLP printing technologies use a projector with conventional light to cure the whole layer of the built surface with one flash. The resolution of the printed layer has been set to 0.01 mm, so that the precision on the *z* axis is as high as possible for the 3D-printed parts. Our reference was the worm wheel that is shown in Figure 9a. The geometric details (Figure 9b) are so small that the only technology that was found to be suitable was the DLP (Figure 4). Also, because of the dimensions of the teeth of the worm wheel, the only way in which the part could be printed was parallel the XY plane of the print platform.

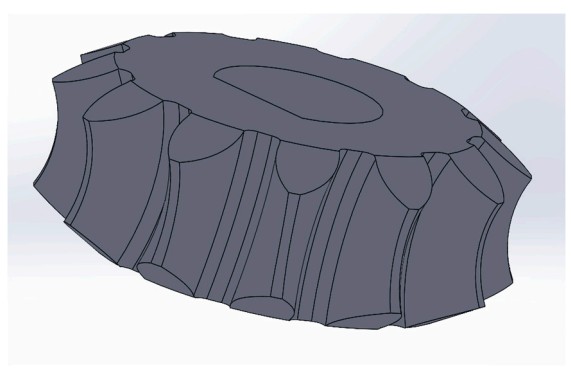

(**a**)

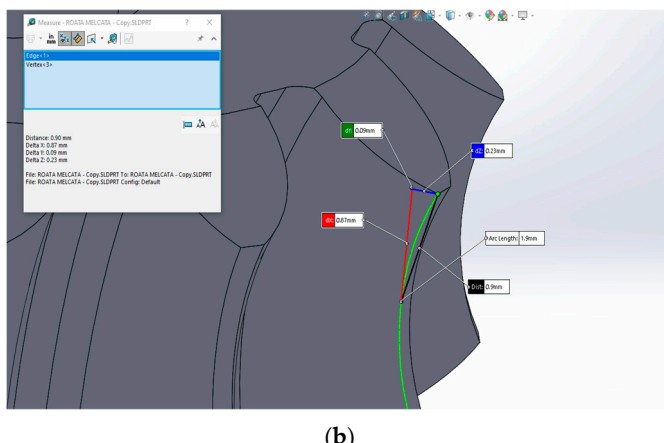

(**b**)

**Figure 9.** Worm wheel geometric details: (**a**) worm wheel—geometric part reference; (**b**) geometric details—part 1.

There were used two types of materials (resins) with this DLP technology, one is TOUGH-GRY 10 and the other is PRO-BLK 10 (see Table 1). In addition to the mechanical characteristics described in the table below, there were a couple of additional aspects to note, as follows:

- The material TOUGH-GRY 10 is a fast-printing engineering material (as noted from observation and experience with printing), and we could use it to print parts that do not require high geometrical accuracy.
- The material PRO-BLK 10 is a material that prints parts with high accuracy parts, the speed is reduced compared to that of TOUGH-GRY 10.

- PRO-BLK 10 is designed to require less support than any other engineering materials used so it is easier to clean, cure, and postprocess parts that need high accuracy.

**Table 1.** Material characteristics.

|  | **TOUGH-GRY 10** | **PRO-BLK 10** |
|---|---|---|
| Tensile Strength Ultimate | 50 MPa | 63 MPa |
| Elongation at Break | 25% | 12% |
| Flexural Strength | 75 MPa | 92 MPa |

The printer used was "Figure 4 Standalone", from 3D Systems [20]. Detailed descriptions of the printing processes for the hand prosthesis components are provided in [21].

The printing parameters used (print settings and support settings) are presented in Table 2.

**Table 2.** Printing parameters.

| **Material Settings** | **Support (Gate Type)** | |
|---|---|---|
| Layer height: 10 microns<br>Print mode: premium build<br>(moderate speed; best print quality) | Tip | Penetration length: 0.5 mm<br>Pillar top ration: 0.275<br>Pillar top height: 1 mm<br>Pillar bottom height: 1 mm |
| | Bridge | Longest bridge length: 5 mm<br>Min. dis. Bridge–Bridge: 1.354 mm |
| | Pillar | Chunk Pillar Width: 0.4 mm<br>Pillar offset Upper Bound: 6.768 mm<br>Pixel–Boundary Distance: 0.203 mm |
| | Truss | Truss Z interval: 3 mm<br>Truss Thickness Ratio: 0.8<br>Longest Truss: 10 mm |

The SLS printing process builds parts by embedding them in a powder bed, which means that the parts are printed without the help of a support, as in the case of other technologies. The powder that is not sintered by the printer laser acts as a support for the slice geometry that is built.

For the prototyping prosthesis components, a Fuse 1 [19] printer was used; in this process, we additionally utilized the Preform 3.28.1—January 2023 software for the slicing process and thus obtained adequate layers for accurately printed parts. The material used for the printed parts was Nylon12, with a 30% refresh rate (30% new powder, 70% reclaimed powder). Nylon has some very interesting properties for prototyping and for series production:

- It bends and returns to its original form rather than experiencing sudden fracture, unlike ABS or PLA;
- It is excellent for structural, load-bearing, or mechanical parts;
- It is suitable for high-temperature applications.

The printing parameter that the user can change or adjust is the refresh ratio; however, because we externalized the printing of the components, the refresh rate was set to the default, which was 30% of the new material.

Because the here-designed hand prosthesis is, at this stage, in preliminary alpha testing, we did not consider the usage of biocompatible materials. For now, testing was performed to assess the functionality with regard to the design and programming of the controllers. Moreover, the hand parts of the prosthesis do not come into direct contact with the body, so the biocompatibility for these components is not a priority in the present research.

We will consider the technology and 3D printers available in order to print with biocompatible materials when the time comes. In that regard, the Figure 4 3D printer has the capacity to print with two biocompatible materials (MED-AMB 10, MED-WHT 10). In fact, according to the material filter on the 3D Systems website, the material PRO-BLK 10 which we used to print the parts appears to be adaptable for biocompatibility [21].

For the Fuse 1 printer, the nylon 12 material used to print some of the hand prosthesis parts is considered to be biocompatible, based on producer tests [20].

To choose the best 3D-printing technology for our case study, a decision matrix was designed (see Table 3). The matrix takes into account multiple printing parameters, like setup flexibility, accuracy, repeatability, build quality, etc. The criteria are ranked from 1 to 10, with 10 being the best option.

**Table 3.** Decision matrix.

| Printer<br>Criteria | Figure 4 | MarkTwo | Form 2 | Fuse 1 | Jet Fusion 5200 | Weighting |
|---|---|---|---|---|---|---|
| Setup flexibility | 6 | 8 | 6 | 5 | 4 | 12% |
| Build quality | 9 | 8 | 6 | 8 | 8 | 22% |
| Mechanical processing | 8 | 3 | 6 | 8 | 7 | 8% |
| Accuracy | 9 | 8 | 7 | 8 | 7 | 20% |
| Repeatability | 9 | 9 | 8 | 9 | 9 | 6% |
| Speed | 10 | 5 | 5 | 7 | 6 | 5% |
| Volume | 4 | 6 | 4 | 7 | 9 | 7% |
| Cost of manufacturing | 8 | 5 | 5 | 9 | 7 | 20% |
| Total | 8.06 | 6.77 | 5.93 | 7.78 | 7.07 | 100% |

The criteria used to analyse the results are explained bellow:

- Setup flexibility—Can the parts be rearranged in the printer volume with ease, if the parts are modified by the end user? This can be judged by the number of hours/minutes the designer needs to rearrange or modify the build after part modifications.
- Build quality—How long will the parts last in use? This can be judged based on the quality of the materials used for the 3D print.
- Mechanical processing—Is the design easy to mechanically readjust after the print? Measured by the ease of parts modification by the user in regard to mechanical processing.
- Accuracy—Does the printed part accurately meet the geometry of the CAD design? Measured through the tolerances of the finished part.
- Repeatability—Can the 3D printer achieve the same output when printing the same build? Measured by accuracy of the same parts printed in different builds.
- Speed—How much time does the 3D printer take to finish a print job and start the next?
- Volume—How big is the 3D-printing space (volume)?
- Cost of manufacturing—How much do the parts cost to manufacture?

Based on the decision matrix results, the best option for manufacturing the parts was the Figure 4 printer from 3D Systems. Alongside this, the Fuse 1 printer was chosen for printing some of the bigger parts due to the high build capacity and the reduced cost for our required volume of parts.

## 3. Results and Discussion

Designing the hand prosthesis has been a difficult process because of the following requirements:

- The prototype size and aspect must be similar to those of a real hand;
- The mechanical parts are small (for example, module of 0.06 mm for gears) and require high precision (for example, 0.005 and 0.01 mm tolerances);

- The mechatronic components have reduced dimensions and mass;
- Low weight and user-friendly interface;
- Relatively affordable price.

There were different variants considered for the mechanical structure of the fingers, depending on the number and position of levers and joints to ensure the correct motion of the fingertips. Importantly, we considered the type and characteristics of the wheel gears and/or the wheel–worm gears, the thread pitch dimensions, the available space for mounting the components (because of physical limitation; resulted from reverse engineering technique). Calculating and deciding on these different elements were considered to be milestones for obtaining the physical prototype.

The model for the hand prosthesis is presented in Figure 10.

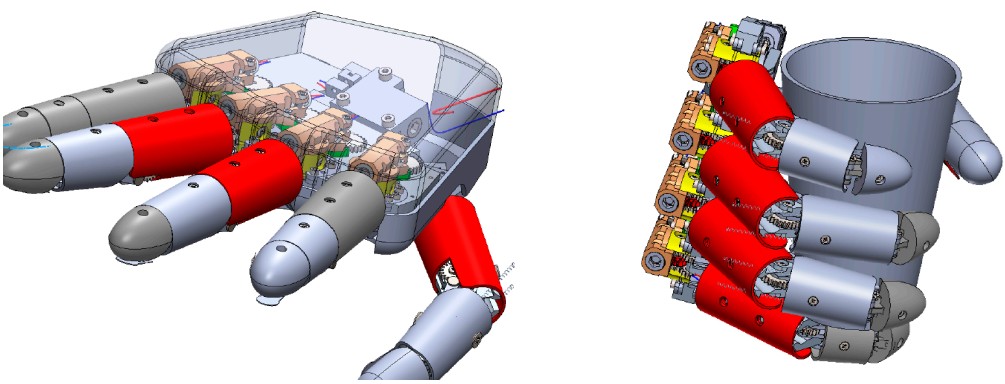

**Figure 10.** Hand prosthesis 3D model.

Kinematic analysis for the fingers was performed as described in Section 2 (see Figure 3). The numerical results and plotted graphs obtained for the angular position of the kinematic elements ($\varphi_1 \div \varphi_4$) and for the coordinates of the representative finger points (A, D, T) are shown in Tables 4 and 5, respectively.

**Table 4.** Values for finger phalanges angles.

| Incremental Position | $\varphi1$ | $\varphi2$ | $\varphi3$ | $\varphi4$ | [mm] |
|---|---|---|---|---|---|
| 0 | 3.2000 | 3.3000 | 0.9690 | 2.9600 | |
| 1 | 3.2175 | 3.3524 | 1.0403 | 3.0062 | |
| 2 | 3.2349 | 3.4047 | 1.1119 | 3.0526 | |
| 3 | 3.2524 | 3.4571 | 1.1839 | 3.0992 | |
| 4 | 3.2698 | 3.5094 | 1.2561 | 3.1460 | |
| 5 | 3.2873 | 3.5618 | 1.3287 | 3.1930 | |
| 6 | 3.3047 | 3.6142 | 1.4017 | 3.2403 | |
| 7 | 3.3222 | 3.6665 | 1.4749 | 3.2878 | |
| 8 | 3.3396 | 3.7189 | 1.5484 | 3.3356 | |
| 9 | 3.3571 | 3.7712 | 1.6223 | 3.3836 | |
| 10 | 3.3745 | 3.8236 | 1.6964 | 3.4320 | |
| 11 | 3.3920 | 3.8760 | 1.7708 | 3.4806 | |
| 12 | 3.4094 | 3.9283 | 1.8455 | 3.5295 | |
| 13 | 3.4269 | 3.9807 | 1.9204 | 3.5788 | |
| 14 | 3.4443 | 4.0330 | 1.9958 | 3.6283 | |
| 15 | 3.4618 | 4.0854 | 2.0713 | 3.6782 | |

*Observation*:
Incremental positions refer to:
26 equidistant position of element 1, considering the initial position: $\varphi_1 = \varphi_{10} = 3.2$ rad.

**Table 4.** *Cont.*

| Incremental Position | φ1 | φ2 | φ3 | φ4 | [mm] |
|---|---|---|---|---|---|
| 16 | 3.4793 | 4.1378 | 2.1471 | 3.7284 | |
| 17 | 3.4967 | 4.1901 | 2.2231 | 3.7790 | |
| 18 | 3.5142 | 4.2425 | 2.2994 | 3.8298 | |
| 19 | 3.5316 | 4.2948 | 2.3759 | 3.8811 | |
| 20 | 3.5491 | 4.3472 | 2.4527 | 3.9327 | |
| 21 | 3.5665 | 4.3996 | 2.5296 | 3.9846 | |
| 22 | 3.5840 | 4.4519 | 2.6068 | 4.0369 | |
| 23 | 3.6014 | 4.5043 | 2.6841 | 4.0896 | |
| 24 | 3.6189 | 4.5566 | 2.7617 | 4.1427 | |
| 25 | 3.6363 | 4.6090 | 2.8393 | 4.1961 | |

**Table 5.** Coordinates values for A, D, and T phalanges joints.

| Incremental Position | XA | YA | XD | YD | XT | YT | [mm] |
|---|---|---|---|---|---|---|---|
| 0 | 0.0670 | 0.0425 | 0.0386 | 0.0398 | 0.0187 | 0.0379 | |
| 1 | 0.0670 | 0.0419 | 0.0389 | 0.0374 | 0.0191 | 0.0348 | |
| 2 | 0.0671 | 0.0410 | 0.0393 | 0.0350 | 0.0197 | 0.0310 | |
| 3 | 0.0671 | 0.0401 | 0.0397 | 0.0327 | 0.0205 | 0.0273 | |
| 4 | 0.0672 | 0.0391 | 0.0402 | 0.0303 | 0.0214 | 0.0235 | |
| 5 | 0.0673 | 0.0382 | 0.0408 | 0.0280 | 0.0226 | 0.0199 | |
| 6 | 0.0675 | 0.0373 | 0.0415 | 0.0257 | 0.0239 | 0.0163 | |
| 7 | 0.0676 | 0.0364 | 0.0423 | 0.0234 | 0.0254 | 0.0128 | |
| 8 | 0.0678 | 0.0355 | 0.0432 | 0.0212 | 0.0271 | 0.0093 | |
| 9 | 0.0679 | 0.0346 | 0.0441 | 0.0191 | 0.0290 | 0.0060 | |
| 10 | 0.0681 | 0.0336 | 0.0452 | 0.0169 | 0.0310 | 0.0028 | |
| 11 | 0.0683 | 0.0327 | 0.0463 | 0.0148 | 0.0332 | −0.0003 | |
| 12 | 0.0686 | 0.0318 | 0.0475 | 0.0128 | 0.0356 | −0.0033 | |
| 13 | 0.0688 | 0.0309 | 0.0487 | 0.0108 | 0.0381 | −0.0061 | |
| 14 | 0.0690 | 0.0301 | 0.0501 | 0.0089 | 0.0407 | −0.0087 | |
| 15 | 0.0693 | 0.0292 | 0.0515 | 0.0071 | 0.0434 | −0.0112 | |
| 16 | 0.0696 | 0.0283 | 0.0529 | 0.0053 | 0.0463 | −0.0136 | |
| 17 | 0.0699 | 0.0274 | 0.0545 | 0.0036 | 0.0493 | −0.0158 | |
| 18 | 0.0702 | 0.0265 | 0.0560 | 0.0019 | 0.0524 | −0.0177 | |
| 19 | 0.0705 | 0.0257 | 0.0577 | 0.0003 | 0.0555 | −0.0195 | |
| 20 | 0.0709 | 0.0248 | 0.0594 | −0.0011 | 0.0587 | −0.0211 | |
| 21 | 0.0712 | 0.0240 | 0.0611 | −0.0026 | 0.0620 | −0.0226 | |
| 22 | 0.0716 | 0.0231 | 0.0629 | −0.0039 | 0.0653 | −0.0238 | |
| 23 | 0.0720 | 0.0223 | 0.0647 | −0.0052 | 0.0686 | −0.0248 | |
| 24 | 0.0724 | 0.0214 | 0.0665 | −0.0064 | 0.0720 | −0.0256 | |
| 25 | 0.0728 | 0.0206 | 0.0684 | −0.0074 | 0.0753 | −0.0262 | |

*Observation*:
Incremental positions refer to:
26 equidistant position of element 1, considering the initial position: $\varphi_1 = \varphi_{10} = 3.2$ rad.

Because the hand prosthesis, for the time being, is intended to be a prototype, we utilised rapid prototyping manufacturing technologies [22].

One such technology was SLS printing on a Fuse 1 printer. The disadvantage of this technology is that some of the obtained parts (Figure 11) did not attain the required geometric precision.

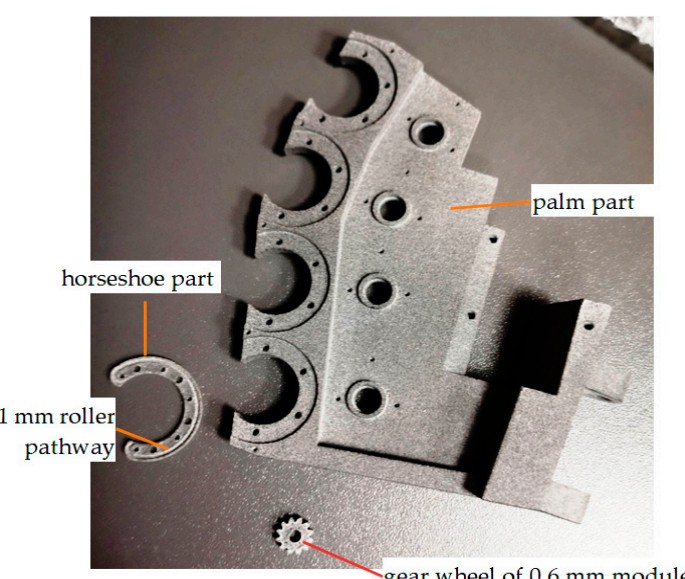

**Figure 11.** Fuse 1-printed components.

A relevant example is that of the wheel gears, whose module resulted in a slightly higher value than the required 0.6 mm. Thus, motion transmission to the associated gear wheel could not be correct.

Another example is that of the horseshoe part, which had the correct outer diameter, enabling it to be correctly assembled with the "palm" part; however, the 1 mm roller pathway, for the rollers of the bearings, could not be adequately prototyped—the 1 mm dimension was not constant for the full pathway length, and the surface roughness was so high that the roller process was, instead, more of a sliding process, which is unacceptable in a bearing system.

The Figure 4 printer used a DLP printing technology. After curing the parts with UV light, the components turned from "green parts" into final parts, which were ready to be assembled into the hand prosthesis's subsystems (see Figure 12).

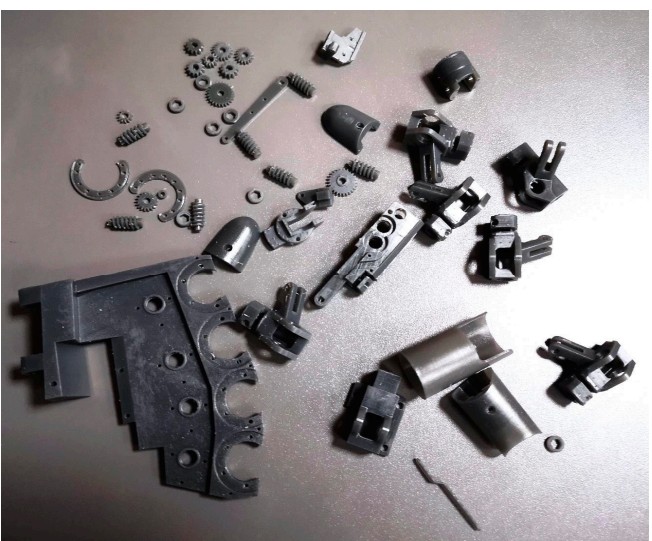

**Figure 12. Figure** 4-printed components.

The printed parts must be assembled with all the other components (roller bearing, micromotors, sensors) in order to test the prosthesis. In order to avoid the eventual loss of time and material (because of inadequate printing), we decided to assemble and test two fingers first: the thumb and the index finger. It was determined that, if those two fingers worked correctly, then the other three fingers would be assembled in order to obtain the complete prototype of the hand prosthesis.

For the parts that did not need tolerance values lower than 0.05 mm, there were no problems in assembling them into subsystems: see, for example, the thumb shown in Figure 13.

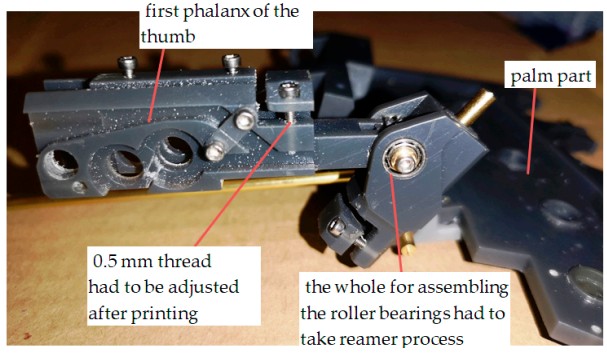
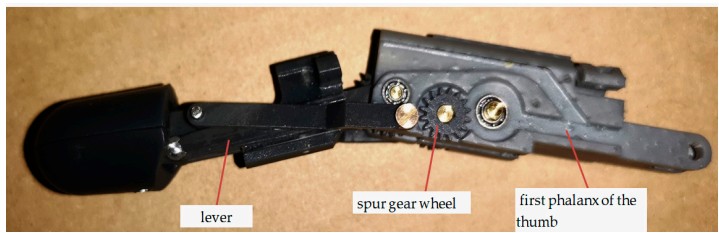

**Figure 13.** Thumb prosthesis subassembly.

Tests of the micromotors for the spur cylindrical gear were performed. The first experiment provided the correct result for the motion; however, repeated testing resulted in findings of incorrect finger motion. This happened because the grey printed material (TOUGH-GRY 10) does not offer suitable wear resistance and the teeth of the wheel in the worm–wheel gear becomes worn out quickly (see Figure 14). This is why it was important to perform many testing phases using different materials and adjustments to the gear wheels.

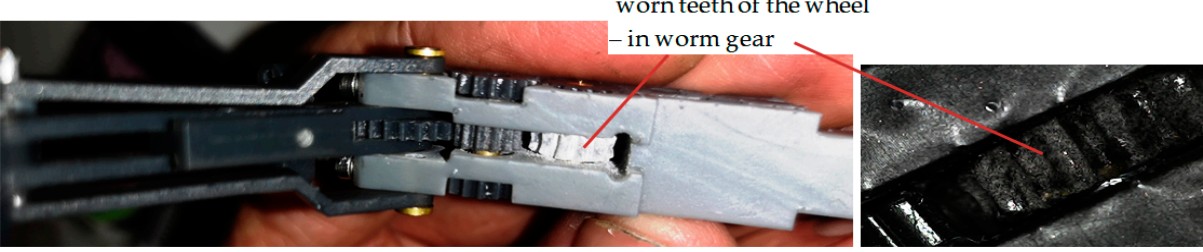

**Figure 14.** Thumb prosthesis functional tests—teeth worn out.

An image of the process of testing the prosthesis prototype's finger movement, fixture, and control system is shown in Figure 15. The obtained results for the motions of two fingers (the thumb and the index finger) were adequate; therefore, the whole prototype of the hand prosthesis seemed to work properly according to its concept and design. Due to the fact that the other three fingers (middle, ring, and little) are very similar in structure to the index finger, and the movement of the index finger was good, we made the sound assumption that everything would function correctly for the other three fingers.

As mentioned in Section 2, a questionnaire on the need and requirements for prostheses was designed and shared through Google Forms for market research. There were more than 50 responses that highlighted a need for hand prostheses among people with a missing upper limb (Figure 16).

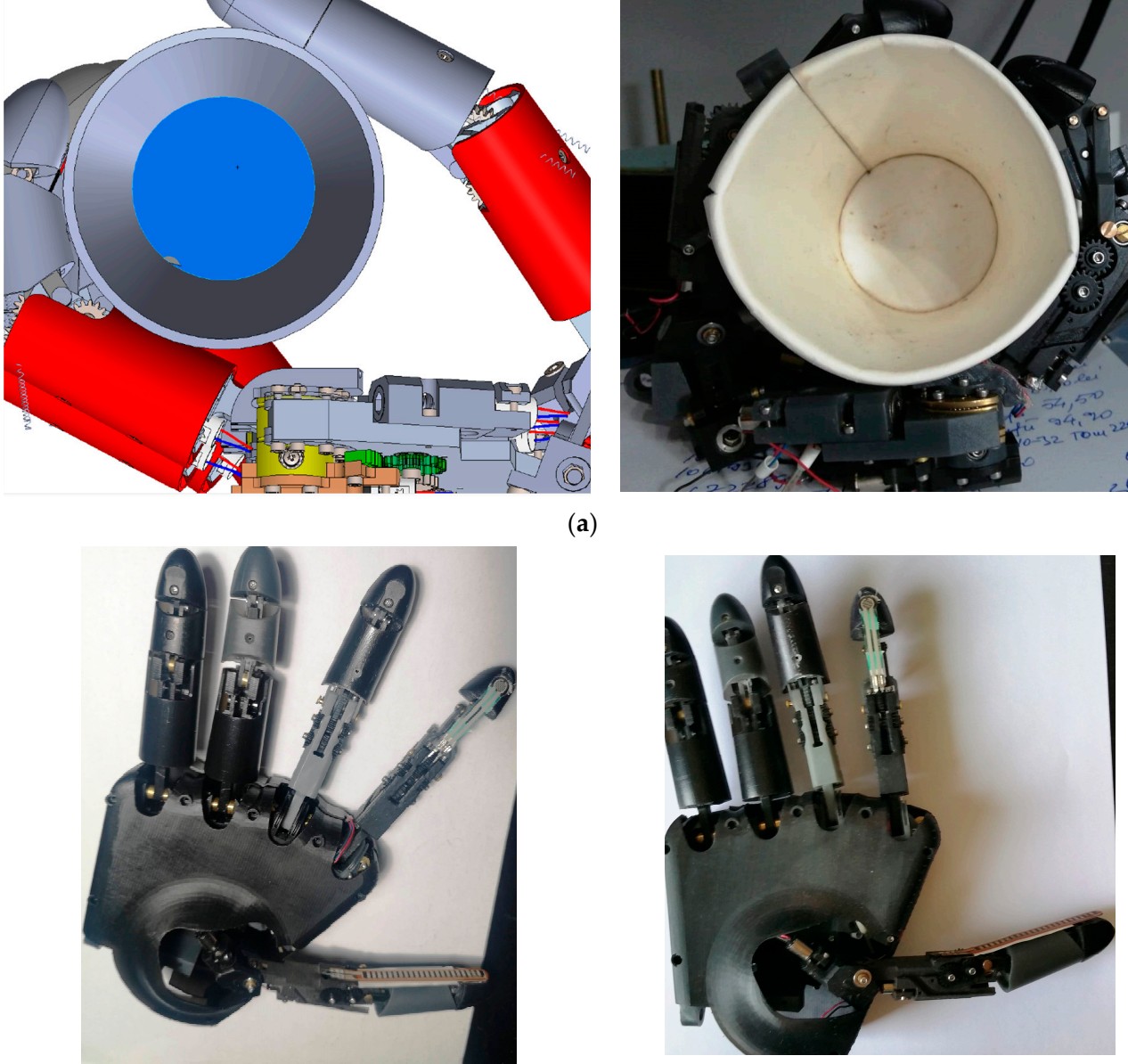

**Figure 15.** Hand prosthesis prototype: (**a**) grabs object with a cylindrical surface; (**b**) rotation of index and middle fingers.

Most of the responders stated the need for a hand prosthesis because it would enable them to have a relatively "normal" life. The need for governmental support was also mentioned with respect to the need for financially affordable prostheses.

Focusing on question number 17 (related to the need for higher-performance prostheses), the respondents were asked to rank, by marks from 1 to 5, the requirements it should meet. The results are shown in Table 6. There are some significant points to highlight here:

- The marks shown in Table 6 are averages of the marks awarded by each of the 45 respondents (who stated the need for a higher-performance prosthesis);
- The weight values have been assigned based on the manufacturing and use conditions for the prosthesis developed in this study.

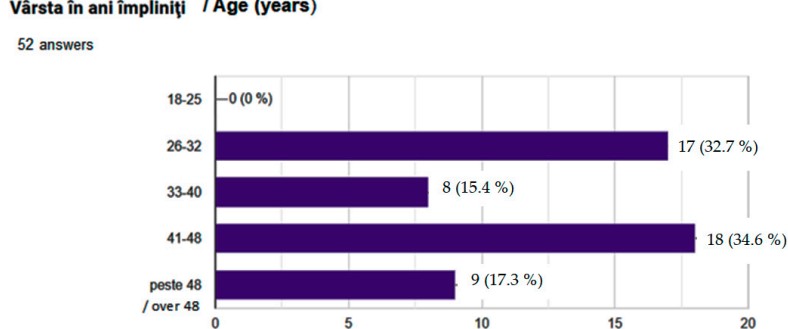

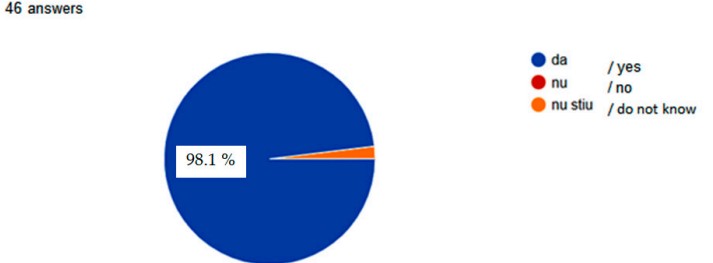

**Figure 16.** Questionnaire answers on the need for hand prostheses.

**Table 6.** Volunteer respondents ranks for prosthesis requirements.

| Requirements | Mark | Weight | Results (Mark × Weight) |
|---|---|---|---|
| Price | 4.6 | 0.23 | 1.058 |
| User-Friendly (for use) | 3.5 | 0.26 | 0.910 |
| Movement Complexity | 4 | 0.31 | 1.240 |
| Abutment attach and psychological impact | 4.3 | 0.2 | 0.860 |

The results highlight the need for a hand prosthesis to enable complex motions and therefore improve functionality and quality of life. This was followed by the price element—which is dependent on the social welfare of the respondents who agreed to participate in the survey. Other aspects involved the interaction between the body and the prosthesis (use and psychology); these were rated relatively closely to each other but were considered less important than complexity and affordability.

## 4. Conclusions

This paper presents the basic aspects and the research results obtained in designing an innovative hand prosthesis. This research paper is an extended version of the initial paper, "Basics of Hand Prosthesis Design", that was selected for the conference named "New Trends in Medical and Service Robots, MESROB 2023".

The focus of the paper is on the relevant stages of designing a hand prosthesis prototype that is innovative in its mechanical structure and, therefore, in the degree of freedom and mobility of the fingers. The authors' concept and design for the hand prosthesis are innovative as a result of several factors, as follows: the rotations for each of the phalanges; the additional rotation for the whole finger; the independent motions for the each of the fingers; the mechanical components generate finger motion based on levers and gears (spur, worm) with no cables. The hand prosthesis is a complex system with embedded command and control.

The reverse engineering technique applied to determine the dimensions of a person's existing arm in order to tailor-make their prosthesis is an original approach to ensuring that the outside surface has the dimensions of the person's real hand.

The manufacturing techniques considered were prototyping by additive manufacturing, specifically: SLS printing on a Fuse 1 printer and DLP printing on a Figure 4 printer. Because of specific problems caused by the small dimensions (module of 0.06 mm for gears) and high precision (0.005 and 0.01 mm tolerances) required for the mechanical parts, different available (for this research) prototyping techniques had to be tested.

A questionnaire for market research on the need and requirements for prostheses was designed and shared using Google forms. The results pointed out the need for the hand prosthesis to enable complex motions and therefore enable a relatively "normal" life. This was followed closely by the aspect of affordability—which is dependent on the social welfare of the volunteers who agreed to participate in the survey.

The conclusion made by the authors regarding this innovative hand prosthesis design is that the results could be extended to prototyping customized prostheses for people with missing limbs in a series of specific steps, as follows:

- The reverse engineering technique represents innovative approach: it saves time and effort in ensuring a good aesthetic appearance for the prosthesis and in determining the correct exterior surface dimensions;
- The innovative mechanical system components for the motions of the fingers are to be scaled according to the exterior surface dimensions of the prosthesis;
- The embedded command and control system is designed according to the mechanical structure of the prosthesis using a versatile hardware platform; these elements can be customized for the specific requirements of the person the prosthesis is being made for (haptic, AI guidance, etc.);
- The higher-performance additive manufacturing technologies could be applied for prototyping (like laser melting of metal-based alloy powders) the mechanical components, thus reducing the time required for assembly.

There are, as any product has, limitations of this prosthesis's performance: the type of actions to be performed (for example, the prosthesis may experience difficulty in cutting food pieces, combing hair, grabbing heavy objects, etc); the energy consumption required (depending on available market offers and battery weight); the costs for manufacturing.

Attention should be given to the way the prosthesis is to be attached to person's abutment—this aspect has not yet been directly addressed in this study.

Further research developments will focus on testing the hand prosthesis and, if required, adjusting it in accordance with the feedback provided by the questionnaire participants.

**Author Contributions:** Conceptualization, E.F.L. and M.I.; methodology, C.R., M.M.R., and L.M.; software, L.M.; validation, E.F.L., C.R., and M.I; investigation, E.F.L., M.M.R., and L.M.; data curation, L.M. and M.I.; writing—original draft preparation, C.R. and M.M.R.; writing—review and editing, M.I. and L.M.; visualization, E.F.L.; supervision, M.I. All authors have read and agreed to the published version of the manuscript.

**Funding:** This research received no external funding.

**Institutional Review Board Statement:** Not applicable. Ethical review and approval were waived for this study due to the fact that all research involved only the authors.

**Informed Consent Statement:** Informed consent was obtained from all subjects involved in the study, as it involved only the authors.

**Data Availability Statement:** Not applicable.

**Acknowledgments:** We specially thank CADWORKS International SRL for the support and help given in the elaboration of this work. *** https://www.cadworks.ro/ (accessed on 20 December 2022) ***. We also thank the Romanian Academy for the support of this research.

**Conflicts of Interest:** The authors declare no conflict of interest.

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
