# Peer review of "Biomechanical Hand Prosthesis Design†"

_machines, doi:10.3390/machines11100964_

Round 1

Reviewer 1 Report (Previous Reviewer 1)

The paper presents the design of a biomechanical hand prothesis.

 The paper has been considerably improved since the last version, however there are some aspects that may improve the quality of the presentation:

The kinematic analysis is still not detailed, the authors present a few lines of code and reference some methods, but there are no further details, maybe the authors could extend this section with proper detail regarding the kinematics of the hand prothesis.

The functional validation of the hand prothesis is not caried on (or not explained in the text). Since the authors present some figures with the manufactured prototype and control system maybe some data regarding some experimental tests (in lab) would complete the research.

Best regards.

Some sentences could be rephrased :"innovative by its mechanical structure and, therefore, fingers DOF and mobility" ---innovative by increased Degrees of freedom  and mobility for the fingers (just a sugestion)

Author Response

The paper presents the design of a biomechanical hand prothesis.

 The paper has been considerably improved since the last version, however there are some aspects that may improve the quality of the presentation:

The kinematic analysis is still not detailed, the authors present a few lines of code and reference some methods, but there are no further details, maybe the authors could extend this section with proper detail regarding the kinematics of the hand prothesis.

The kinematic analysis mentioned in this paper is forward kinematic one, done classically starting from the active elements (motor rotation in joint H and first mobile element, 1). There are specific procedures designed and available (by contacting the authors of mentioned References [19, 20]). The background of these procedures stands in vector loop method. (lines 243,244)

We used these procedures as it was more convenient to apply computer aided techniques rather than performing step by step the geometric and trigonometric calculi.

In order to apply these procedures, we defined, each of the mechanism elements geometric characteristics (lengths, angles). These are to be noticed in the detailed calculi program but, for data protection, in this paper we provided just a “sample” of it (“few lines of code”).

The functional validation of the hand prothesis is not caried on (or not explained in the text). Since the authors present some figures with the manufactured prototype and control system maybe some data regarding some experimental tests (in lab) would complete the research.

Tests were done from the point of view of fingers movement, fixture and control system. Also, there is the explanation of the reason for these mentioned experiments (lines 493 - 499, 500, 501).

The authors are aware that movies are more suggestive but because this research has as background doctoral research which is under evaluation by accredited commission at the Ministry of Education (in Romania) there are only images to be presented.

Further research development is mentioned at (lines 586, 587)

Best regards.

The authors thank Reviewer for comments and suggestions.

The changes have been highlighted in red colour.

Reviewer 2 Report (Previous Reviewer 2)

This article aims to introduce the innovation of mechanical structures, and there are several questions that the author needs to consider.

(1)    How does the structure stand out among most prosthetic hands? Although an investigation report is provided, can the author provide comparative experiments between the proposed structure and other mechanisms?

(2)    Experiments are conducted on the proposed mechanism in the paper, but there is no data or report provided, making it difficult to add credibility to this structure.

Author Response

This article aims to introduce the innovation of mechanical structures, and there are several questions that the author needs to consider.

  • How does the structure stand out among most prosthetic hands? Although an investigation report is provided, can the author provide comparative experiments between the proposed structure and other mechanisms?

At (lines 211 - 221) there are mentioned innovative aspects of this prosthesis design like: rotations for each of the phalanges; independent motions for the each finger;  motions achieved by levers and gears (spur, worm) with no cables and elastic elements (spring);  command and control – by micromotors (not servomotors and encoders), microcontroller and sensors’ signals, etc.

The authors carefully studied the research articles (as evidenced in the References) and the movies / data available on social media - all focused on high performance prosthesis.

At the beginning of this research, the authors attended a workshop at a Romanian SME (https://www.activ-ortopedic.ro/) which is specialised in leg prosthetic but a team from Open Bionics had a demonstration on the thematic of  hand prosthesis.

Other similar prosthesis could not be (really) tested by the authors, as none of the people involved in the statistic study (market research on the need and requirements for prostheses) had a good performance hand prosthesis in use.

(2)    Experiments are conducted on the proposed mechanism in the paper, but there is no data or report provided, making it difficult to add credibility to this structure.

Tests were done from the point of view of fingers movement, fixture and control system. Also, there is the explanation of the reason for these mentioned experiments (lines 493 - 499, 500, 501).

The authors are aware that movies are more suggestive but because this research has as background doctoral research which is under evaluation by accredited commission at the Ministry of Education (in Romania) there are only images to be presented.

Further research development is mentioned at (lines 586, 587)

Credibility of the structure - sounds awkward, as this article presents research results at a certain stage of the product development process.

There can also happen failure in research - which are also worth to be published so that to other researchers could avoid those paths.

Still, based on experiments presented in this article (lines 473 - 478, 484 - 490, 493 - 499) and on many others that are not available for publication yet, the authors are confident that the concept and design  of this hand prototype will be of benefit for many poor / middle class (young) people.

The authors thank Reviewer for comments and suggestions.

The changes have been highlighted in red colour

This manuscript is a resubmission of an earlier submission. The following is a list of the peer review reports and author responses from that submission.

Round 1

Reviewer 1 Report

The paper presents an extended research on biomechanical hand prosthesis design.

The work presented is of interest however there are some suggestions that may improve the quality of the paper:

Introduction

-the first part of the introduction is more like a story and it lacks the scientific perspective, the authors expose some of types of prosthesis used since ancient time however the data should be sustained with scientific work, for example the first reference cited in the paper is on the 12th paragraph of the introduction.

- the introduction end with a paragraph presenting the aim of the paper, however this aim should be written in such manner to highlight the innovative aspect of the research, what is the novelty of this research? What is different than the other presented research?

-the introduction may end with a short summary of the paper where the chronological sections should be briefly explained (e.g. “the second section presents the methods and materials where the design of the hand prosthesis is presented”…)

Materials and methods

-in line 176 use some notations that are not presented H(0R1), what represent the numbers in the brackets?

-in line 206, maybe the link for the  google form could be shortened , for example use a hyperlink for a keyword.

-the authors perform kinematic analysis, is there a kinematic model developed ?

Results and Discussions

-in line 323 is written 00.6 mm

-figure 11 is not readable

The number of references should be extended.

Best regards

Some phrases should be rephrased in a more scientific manner. 

Reviewer 2 Report

1.      Each symbol in Figure 3(b) needs to be explained in terms of its meaning.

2.      The focus of the article is on 3D printing materials, and the simulation experiments have confirmed the feasibility of the material. Although the advantages and disadvantages of various materials are listed earlier, the subsequent simulation does not compare the practicality of the proposed material.

3.      The emphasis in the introduction section does not accurately reflect the focus of the main body. Please correct the introduction section.

4.      The article mentions the use of two resins (TOUGH-GRY 10 and PRO-BLK 10) for 3D printing materials. Please provide a detailed explanation of why these two resins were used and their advantages.

5.      The article also includes a survey questionnaire, but this is not detailed enough and its relevance to the article is weak. It is recommended to accurately classify the needs or purposes of the volunteers, create a table, and provide a detailed explanation of the relevance or purpose of the survey in relation to this design.

 Minor editing of English language required

Reviewer 3 Report

The main topic of the submitted manuscript deals with the development of a new prototype of hand prosthesis. Despite the current interest in innovative robotic devices, the manuscript needs fundamental revisions and it is unclear the innovation and advantages of the proposed model compared to previous prototypes. Here some general suggestions:

The introduction needs a deeper revision. Some parts (historical overview) can be omitted, while additional information of current hand prosthesis models and prototypes must be depicted. It i important to describe researches of the last years in terms of innovations, advantages, results and limitations. Add meaningful citations. The innovation of the current research is not clear.

Methodology section is confused, it is not clear which is the most important aspect of the design (mechanical? control? adopted materials?). It seems that the prototype has been developed for individual need: how it can be adopted for larger usage? It is unclear the connection between the questionnaire and the development of the prototype.

Kinematic analysis must be described in details, Figure 5 (programming code) is not useful.

The control scheme must be deeper explained and it is part of the methodoloogy, not results. Similarly, the assembly cannot be considered as results. Kinematic coordinates need deeper explanations and units must be reported in table.

Results and discussion do not support the study. They must be completely revised. Limitations of the current study must be reported.

No comments about English form, just check possible spelling errors.